# The Emerging Therapeutic Landscape of ALK Inhibitors in Non-Small Cell Lung Cancer

**DOI:** 10.3390/ph13120474

**Published:** 2020-12-18

**Authors:** Valerio Gristina, Maria La Mantia, Federica Iacono, Antonio Galvano, Antonio Russo, Viviana Bazan

**Affiliations:** 1Department of Surgical, Oncological and Oral Sciences, University of Palermo, 90127 Palermo, Italy; valerio.gristina@you.unipa.it (V.G.); maria.lamantia@you.unipa.it (M.L.M.); federica.iacono@you.unipa.it (F.I.); antonio.galvano@unipa.it (A.G.); antonio.russo@usa.net (A.R.); 2Department of Biomedicine, Neuroscience and Advanced Diagnostics (Bi.N.D.), Section of Medical Oncology, University of Palermo, 90127 Palermo, Italy

**Keywords:** non-small cell lung cancer (NSCLC), tyrosine kinase inhibitors (TKIs), ALK inhibitors, crizotinib, ceritinib, alectinib, brigatinib, lorlatinib, ensartinib

## Abstract

The treatment of metastatic non-small cell lung cancer (NSCLC) has undergone a paradigm shift over the last decade. Better molecular characterization of the disease has led to the rapid improvement of personalized medicine and the prompt delivery of targeted therapies to patients with NSCLC. The discovery of the EML4-ALK fusion gene in a limited subset of patients affected by NSCLC and the subsequent clinical development of crizotinib in 2011 has been an impressive milestone in lung cancer research. Unfortunately, acquired resistances regularly develop, hence disease progression occurs. Afterward, modern tyrosine kinase inhibitors (TKIs), such as ceritinib, alectinib, brigatinib, and lorlatinib, have been approved by the Food and Drug Administration (FDA) for the management of anaplastic lymphoma kinase (ALK)-positive NSCLCs. Several compounds are currently under investigation to achieve the optimal strategy of therapy. Additionally, the results of ongoing clinical trials with novel-generation TKI will provide more evidence on the best sequence in the treatment of ALK-positive NSCLC patients. In this review, we provide a comprehensive overview of the state-of-the-art targeted therapy options in ALK-positive NSCLCs. Resistance, potential therapeutic strategies to overcome drug resistance, and future perspectives for this subset of patients are critically analyzed and summarized.

## 1. Introduction

Lung cancer is the leading cause of cancer-related death worldwide in both men and women, with <20% 5-year Overall Survival (OS) for newly diagnosed patients [1]. Based on histopathological features, lung cancers are classified into two main groups: non–small cell lung cancer (NSCLC; 80–85%) and small cell lung cancer (15–20%) [2]. NSCLCs are further subcategorized into three main types: adenocarcinoma (50%), squamous-cell carcinoma (30%), and large-cell carcinoma (15%). However, recent evidence suggests that lung cancer represents a group of molecularly heterogeneous diseases even within the same histological subcategory [3]. About 3–5% of patients affected by NSCLC harbor chromosomal rearrangements in the anaplastic lymphoma kinase (ALK) gene [4]. Cancers harboring rearrangements in the ALK gene are susceptible to treatment with tyrosine kinase inhibitors (TKIs), which inhibit downstream signaling pathways, binding to receptor tyrosine kinases.

Anaplastic lymphoma kinase is a member of the insulin receptor protein−tyrosine kinase superfamily, originally described as a nucleophosmin (NPM)-ALK fusion form in an anaplastic large cell lymphoma (ALCL) cell line. The physiological role of ALK has not been thoroughly clarified, yet some evidence has confirmed the regulatory activity of ALK in the development and function of the central and peripheral nervous systems [5]. In 2007, ALK fusion was reported in NSCLC for the first time in a small cohort (7%) of Asian patients [6]. The most common rearrangement results were from an inter-chromosomal inversion in the short arm of chromosome 2, which creates a fusion between the 5′ portion of the echinoderm microtubule-associated protein like-4 (EML4) gene and the 3′ portion of the ALK gene Inv(2)-(p21p23). As a consequence of the activation of the ALK signaling pathway, the fusion gene EML4-ALK with tyrosine kinase function promotes cell proliferation and survival [7].

Notably, more than seven ALK rearrangements have been identified involving various EML4-ALK breakpoints or, exceptionally, other non-EML4 fusion partners. ALK gene aberrations are more common in the adenocarcinoma histological subtype, in never or light smoker young women and are considered to be largely mutually exclusive with genetic mutations in the epidermal growth factor receptor (EGFR) and KRAS.

Remarkably, central nervous system (CNS) metastases are common in this subset of patients. ALK rearrangements might be promptly detected in tumor tissue using fluorescence in situ hybridization (FISH), immunohistochemistry (IHC), reverse transcription-polymerase chain reaction (RT-PCR), or next-generation sequencing (NGS) [8].

## 2. ALK Inhibitors

Standard chemotherapy was used as the first-line therapy before the discovery of the EML4-ALK fusion protein. ALK inhibitors are one of the breakthrough advances in the NSCLC treatment landscape in the last decade and significant effort has been made toward the creation of novel therapeutic ALK-targeting agents. Indeed, the prognosis of ALK-positive NSCLC patients has dramatically improved as a result of the advent of tyrosine kinase inhibitors (TKIs), which have shown activity against ALK rearrangements [9]. Nowadays, the therapeutic landscape of ALK-positive NSCLC involves different highly potent molecules.

The first ALK inhibitor (ALKi) to be used in metastatic NSCLC and approved by the Food and Drug Administration (FDA) was crizotinib, which targets ALK, ROS-1, and c-MET. Afterward, in 2014 a second-generation ALKi, ceritinib, obtained FDA approval for ALK-positive patients after disease progression on crizotinib or for those who were intolerant to it. Based on the results of the randomized phase III ALEX trial, alectinib was approved for treatment-naïve ALK-positive patients. Thereupon, the FDA authorized brigatinib for those patients who had failed prior ALKi treatment. Most recently, in this fast-growing therapeutic landscape, in order to overtake acquired resistance, prolong the control of the disease, and manage CNS disease, several highly potent next-generation ALK TKIs have been developed as lorlatinib and ensartinib.

In this review, we discuss the principal ALKis currently in use in oncological practice, aiming attention at their efficacy, safety, and their role in the optimal care of patients with advanced or metastatic ALK-rearranged NSCLC. A summary of the main ALK inhibitor trials is presented below in Table 1.

### 2.1. Crizotinib

Crizotinib is an oral tyrosine kinase inhibitor (TKI), originally developed as a c-MET inhibitor, approved for the treatment of NSCLC patients with ALK aberration. Crizotinib metabolism mainly involves the CYP3A4/5; thus, hepatic impairment is likely for plasma drug concentrations. There are no clinically significant differences according to age, ethnicity, gender, or bodyweight [10]. Moreover, the impressive activity against ALK-positive cells observed during the dose-escalation portion of the phase I PROFILE 1001 study led to an expansion of the clinical trial. Namely, two patients affected by NSCLC with EML4-ALK rearrangement treated with crizotinib orally 250 mg twice daily showed a remarkable improvement in their symptoms and prolonged objective responses during the dose-escalation phase. Therefore, the clinical trial was expanded, screening and enrolling NSCLC stage III and IV ALK-positive patients in a large cohort treated with the maximum tolerated dose (MTD) of crizotinib 250 mg twice daily in a 28-day cycle. Camidge et al. reported the updated results of the phase I trial, carried out between August 2008 and June 2011, aiming to assess the tolerability and activity of crizotinib in 149 previously treated and untreated ALK-positive NSCLC patients. Of note, 143 patients were included in the response-evaluable cohort. The outcomes demonstrated 3 complete response (CR) and 84 partial response (PR), for a 61% overall response rate (ORR) with a median duration of response (DoR) of 49.1 weeks and a median progression-free survival (PFS) of 9.7 months (95% CI 7.7–12.8). At the time of data cutoff, overall survival (OS) data were not available.

Noteworthy, of the 69 patients who had a documented disease progression, 39 maintained the treatment with crizotinib for more than 2 weeks after progression due to clinical benefit [11,12].

Moreover, the phase II study PROFILE 1005 confirmed these outstanding results on 261 ALK-positive pre-treated NSCLC patients, showing an ORR of 59.8%, with a median duration of response of 45.6 weeks, and a median PFS of 8.1 months (95% CI 6.8–9.7). Owing to the striking and prolonged objective responses observed in both phase I and II trials, in 2011 the U.S. FDA granted approval for crizotinib as the first ALK inhibitor for ALK-positive NSCLC patients. Afterward, two phase III studies (PROFILE 1007 and PROFILE 1014) confirmed the highly potent activity of crizotinib and its superiority to the standard of care of chemotherapy regimens in the second-line and first-line setting, respectively.

Moreover, the multicenter phase III PROFILE 1007 trial compared crizotinib with either pemetrexed or docetaxel in the second-line setting in patients with locally advanced or metastatic ALK-positive NSCLC after disease progression on one prior platinum-based regimen. Median PFS (mPFS), the primary endpoint of the trial, was 7.7 months in the crizotinib group and 3.0 months in the pemetrexed or docetaxel group (for progression or death with crizotinib (HR 0.49; 95% CI 0.37–0.64)). Hence, crizotinib was approved in November 2013 in the second-line setting for ALK-rearranged lung cancer patients after disease progression on platinum doublet therapy.

The promising and striking result of the PROFILE 1007 trial led to the phase III PROFILE 1014 clinical study, which aimed to assess the efficacy of the ALK inhibitor crizotinib compared to standard chemotherapy with pemetrexed plus platinum as the first-line treatment for metastatic ALK-rearranged NSCLC patients. This key study proved crizotinib’s superiority over the standard first-line chemotherapy regimens. As a matter of fact, mPFS was significantly longer in the crizotinib arm (10.9 months) than the standard chemotherapy arm (7 months HR 0.45; 95% CI 0.35–0.60), and the ORR was 74% in the crizotinib arm vs. 45% in the standard chemotherapy arm (*p* < 0.001). However, the trial PROFILE 1014 lacked the use of maintenance pemetrexed in the standard chemotherapy arm and there was an extensive crossover between the two arms, which impaired the molecule’s potential advantage [13,14]. Yet, based on these outcomes, crizotinib became the standard first-line oral TKI agent in patients with ALK-positive metastatic NSCLC. Additionally, the ALK inhibitor crizotinib has shown powerful activity against ROS1 gene rearrangements. ROS1 is a receptor tyrosine kinase of the insulin receptor superfamily and its genetic aberrations have been detected in NSCLC, resulting in cancer cell proliferation and prolonged survival. ROS1 rearrangements are identified in about 1–2% of the NSCLC population, affecting mostly young people, never or light smokers, with adenocarcinoma histology. Thus, in March 2016 crizotinib received the American FDA approval for the treatment of patients with metastatic NSCLC whose tumors are ROS1-positive.

### 2.2. Ceritinib

Ceritinib is a second-generation oral ALK inhibitor which is 20 times as potent as crizotinib, with activity and efficacy against ALK mutations arising after crizotinib exposure, particularly L1196M, G1269A, I1171T, and S1206Y [15,16]. Ceritinib inhibits the autophosphorylation of ALK. Alternative potential targets of ceritinib consist of IGF-1 R, InsR, and ROS1 [17]. The recommended therapeutic dose of ceritinib is 450 mg orally once daily and its metabolism is mainly hepatic through the CYP3A enzyme complex. Ceritinib obtained FDA approval for the treatment of ALK-positive patients who progressed or were intolerant to crizotinib in 2014, and as a first-line therapy in 2017. Approval was based on ASCEND-1 and -2. In fact, the phase I ASCEND-1 trial enrolled 255 locally advanced ALK-rearranged or metastatic NSCLC patients. In the ALK-naïve cohort (*n* = 83), ORR was reported to be 72% and the median DoR was 17 months. In the ALK inhibitor–pretreated patient population (*n* = 163), ORR was noted to be 56% and the median DoR was 8.3 months. mPFS in the ALK inhibitor-naïve patient population was 18.4 months and 6.9 months in patients who had prior exposure to crizotinib [18]. Moreover, in the phase II ASCEND-2 trial, including 140 patients who had received two or more previous treatment regimens (with chemotherapy, one or more platinum doublets), the median DoR was 9.7 months and the mPFS was 5.7 months, comparable with those described in ASCEND-1 [19].

In the subsequent phase III randomized multicenter ASCEND-4 trial, treatment-naïve ALK-positive NSCLC patients were randomized to receive ceritinib or platinum-based chemotherapy until disease progression or unacceptable toxicity. The results demonstrated a mPFS of 16.6 months with ceritinib vs. 8.1 months with standard chemotherapy treatment (HR 0.55; 95% CI 0.42–0.73), and an ORR of 73% in the second-generation ALK inhibitor compared to the chemotherapy arm (27%) [20]. These impressive results were confirmed in the phase III trial ASCEND-5, where patients who progressed on chemotherapy and on crizotinib were randomized to receive ceritinib or chemotherapy as a second-line therapy. mPFS was 5.4 months in the ALK inhibitor arm and 1.6 months in the chemotherapy arm [21]. Notably, no randomized clinical studies have directly compared ceritinib and crizotinib head-to-head, though various meta-analyses across clinical trials have been conducted, suggesting ceritinib to be associated with prolonged PFS and OS compared to crizotinib [6].

### 2.3. Alectinib

Alectinib is a highly potent second-generation ALK and Rearranged during Transfection (RET) gene inhibitor. Alectinib is metabolized by CYP3A4 and it is primarily excreted in feces. Alectinib demonstrated high efficacy against several crizotinib-resistant mutations in ALK, along with L1196M, G1269A, C1156Y, F1174L, 1151Tins, and L1152R but not G1202R [22]. The efficacy of alectinib 600 mg orally twice daily was assessed in two phase II studies conducted in an ALK-rearranged, crizotinib-resistant patient population. In a multicenter single-arm phase I/II trial, Seto et al. aimed to assess the activity of alectinib in Japanese ALK-positive metastatic NSCLC patients, who had received no prior treatment with ALK-TKI. In the phase II part of the study, 46 patients received alectinib at the dose of 300 mg daily and 43 patients achieved an objective response, including two complete responses [23]. In the second pilot study, Shaw et al. demonstrated in a cohort of ALK-rearranged, crizotinib-resistant NSCLC patients an ORR of 48% (95% CI 36–60%) and an mPFS of 8.1 (95% CI 6.2–12.6) months [24]. The outstanding findings of these two trials led to the FDA approval of alectinib for patients with ALK-rearranged, crizotinib-resistant NSCLC in December 2015. Afterward, different clinical trials were conducted in order to evaluate alectinib as a front-line treatment. Therefore, in the phase III J-ALEX study, 207 Japanese patients were randomized to receive alectinib 300 mg orally twice daily or crizotinib 250 mg twice daily. The primary endpoint was PFS. After a median follow-up of 42.2 months in the alectinib arm and 42.4 months in the crizotinib arm, the mPFS was 34.1 months and 10.2 months (HR 0.37) with alectinib and crizotinib, respectively [25]. The second phase III randomized clinical trial, the ALEX trial, was conducted in the Caucasian population with 303 treatment-naïve patients affected by metastatic ALK-positive NSCLC, and compared alectinib with crizotinib in the first-line setting. The primary endpoint was PFS. Striking outcomes were reached, as median PFS was longer with alectinib than with the first-generation ALK inhibitor (34.8 vs. 10.9 months, HR 0.50; 95% CI 0.36–0.70). The ORR was 82.9% with alectinib and 75.5% (*p* = 0.09) with crizotinib. In the intent-to-treat (ITT) population, the HR for disease progression or death was 0.43 (95% CI 0.32–0.58) [26,27]. Notably, there was a dose disparity of alectinib in both ALEX trials, since in the ALEX trial alectinib was administered at a dose of 600 mg twice daily, whereas in the phase III multicentered randomized J-ALEX trial the second-generation ALK inhibitor dose was 300 mg orally twice daily. Nevertheless, the superiority of alectinib over crizotinib was evident, hence in November 2017, the FDA approved alectinib for the first-line treatment of patients with advanced ALK-positive NSCLC, at a recommended dose of 600 mg orally twice per day with food. Furthermore, in the randomized, open-label, phase III clinical trial ALESIA, which enrolled only Asian patients with ALK-positive NSCLC between August 2016 and May 2017, alectinib at a dose of 600 mg orally twice daily was compared to crizotinib as a first-line treatment. The primary endpoint was investigator-assessed PFS. Zhou et al. described that PFS was significantly prolonged with alectinib vs. crizotinib (HR 0.22; 95% CI 0.13–0.38). Moreover, the objective response was 114 out of 125 (91%) with alectinib and 48 out of 62 (77%) with crizotinib, with a longer DoR for alectinib than crizotinib (HR 0.22, 95% CI 0.12–0.40) [28]. Likewise, these data highlight the benefit of alectinib in the first-line setting for ALK-positive NSCLC as reflected in the National Comprehensive Cancer Network (NCCN) guidelines, the European Society for Medical Oncology (ESMO) guidelines, and the Italian Association of Medical Oncology (AIOM) guidelines, which recognize alectinib as a better first-line treatment option than crizotinib [29,30,31].

### 2.4. Brigatinib

Brigatinib is a potent second-generation oral TKI able to inhibit ALK fusions, mutant EGFR (L858R), and ROS1 fusions. Brigatinib displays an inhibitory profile against all 17 known resistance mutations of ALK, including the G1202R and L1196M. In addition, brigatinib has a 12-fold greater potency than crizotinib. Brigatinib is mainly metabolized by CYP2C8 and CYP3A4 and hepatic elimination is the major route of excretion [32]. However, in the phase II ALTA trial, Kim et al. investigated the administration of brigatinib in 222 crizotinib-pretreated locally advanced and metastatic ALK-positive NSCLC patients. A total of 112 patients were randomized to receive brigatinib 90 mg daily, 110 patients received brigatinib 180 mg daily and 11 had a 7-day lead-in at 90 mg. The authors reported that the higher dose of brigatinib had more benefit in terms of efficacy compared to crizotinib. The updated outcomes after 2 years of follow-up were reported by Huber et al.: PFS was 16.7 vs. 9.2 months in the crizotinib arm [33,34]. Following these results, in April 2017 the U.S. FDA granted accelerated approval to brigatinib for the treatment of patients with advanced ALK-positive NSCLC who have progressed on or are intolerant to crizotinib. Based on the results of the phase I/II clinical trials, brigatinib (at a dose of 180 mg once daily with a 7-day lead-in period at 90 mg once daily) was compared to crizotinib in the first-line setting in 275 ALK-positive NSCLC patients in the phase 3 ALTA- 1L trial. The primary endpoint of the study was PFS. The rate of PFS was 24 months with brigatinib and 11 months with crizotinib (HR = 0.49; 95% CI 0.33−0.74). Further, ORR was 71% and 60%, respectively, with brigatinib and crizotinib [35]. Consequently, the FDA recently approved brigatinib for the first-line treatment of patients with ALK-positive metastatic NSCLC in May 2020.

Additionally, the phase III ALTA-3 trial comparing brigatinib and alectinib in locally advanced or metastatic ALK-positive NSCLC patients who have progressed on crizotinib is ongoing [36].

### 2.5. Lorlatinib

Lorlatinib is a third-generation ALK- and ROS1-inhibitor, designed especially to target mutations which drive resistance to crizotinib and next-generation TKIs, including the G1202R solvent-front mutation. In fact, lorlatinib in preclinical models showed a 62-fold increased activity against ALK wild type compared with crizotinib [37]. Further, lorlatinib was developed from crizotinib in order to facilitate CNS penetration and increase survival. Lorlatinib is metabolized mainly by CYP3A4 and UGT1A4, thus the concomitant administration of a strong CYP3A inhibitor should be avoided [38]. In addition, various trials have assessed the efficacy and safety of lorlatinib. Solomon et al. evaluated the overall antitumor activity of lorlatinib in 276 patients with ALK-positive, advanced NSCLC. In this phase II study, patients were enrolled into six different expansion cohorts (EXP1–6) following ALK and ROS-1 mutational status and previous therapy, and received lorlatinib 100 mg orally per day in 21-day cycles. Of note, 112 patients were ALK-positive and 47 patients were ROS1-positive [39]. Overall and intracranial tumor response were the primary endpoints of the study. In treatment-naive patients (EXP1), an objective response was achieved in 27 out of 30 patients (90%; 95% CI 73.5–97.9). In the EXP2–5 cohort, including patients with at least one previous ALK-TKI, objective responses were obtained in 93 out of 198 patients (47%; 95% CI 39.9–54.2). Objective response was reached in 41 out of 59 (69.5%; 95% CI 56.1–80.8) patients who had only received previous crizotinib (EXP2–3A), 9 out of 28 (32.1%; 95% CI 15.9–52.4) patients with one previous non-crizotinib ALK-TKI (EXP3B), and 43 out of 111 (38.7%; 95% CI 29.6–48.5) patients with two or more previous ALK-TKIs (EXP4–5) [40]. Therefore, lorlatinib represents a valid and powerful treatment strategy in heavily pretreated ALK-rearranged NSCLC patients. Based on these results, in November 2018 the American FDA granted accelerated approval to lorlatinib at the dose of 100 mg orally daily for patients with ALK-positive metastatic NSCLC who progressed on crizotinib and at least one other ALK TKI, or for patients who progressed after first-line treatment with alectinib or ceritinib. Latterly, Shaw et al. established that ALK resistance mutations could predict response to lorlatinib. An analysis of baseline plasma DNA using a validated 73-gene cell-free DNA (cfDNA) NGS assay and tumor tissue samples was performed. In a subset of patients who experienced disease progression after one or more second-generation ALK TKIs, ORR was higher among those with ALK mutations detected on plasma DNA or tissue samples, compared to patients without mutations (62% vs. 32% and 69% vs. 27%, respectively). Further, a significantly longer PFS has been reported in patients with ALK mutations detected on tissue genotyping compared to patients without mutations (11.0 vs. 5.4 months; HR 0.47) [41]. Recently, at the ESMO 2020 Annual Meeting, a planned interim analysis of the phase III CROWN trial comparing lorlatinib with crizotinib as a first-line therapy was presented. The open-label randomized multicenter phase III trial involved 296 treatment-naïve patients with ALK-positive locally advanced and metastatic NSCLC, randomized to receive lorlatinib at a dose of 100 mg once daily or crizotinib. The primary endpoint was PFS with a median follow-up for PFS of 18.3 months (95% CI 16.4–20.1) for lorlatinib and 14.8 months (95% CI 12.8–18.4) for crizotinib. Lorlatinib showed a 72% improvement in PFS compared to crizotinib (HR 0.28; 95% CI 0.191–0.413). In addition, lorlatinib was also related to improvements in the best overall response (BOR) rate. A total of 113 (76%) patients receiving lorlatinib achieved a complete response (CR) or a partial response (PR) (*n* = 109). Therefore, lorlatinib resulted in a statistically significant clinical improvement in PFS vs. crizotinib and it should be considered a new first-line treatment option for patients with ALK-positive NSCLC [42,43].

### 2.6. Ensartinib

Ensartinib is a novel second-generation ALK inhibitor created to improve the activity on CNS metastases. This small molecule displayed activity against MET, Axl, ABL, EPHA2, LTK, ROS1, and SLK as well. Preclinical data demonstrated the increased potency of the drug as compared with crizotinib and other II generation TKIs, and ensartinib is capable of inhibiting ALK variants (F1174, C1156Y, L1196M, S1206R, T1151, and G1202R mutants). Horn et al. assessed the administration of ensartinib in a phase I/II trial, which enrolled 37 patients with solid tumors in the dose-escalation part and 60 ALK-positive NSCLC patients in the dose-expansion part. In the dose-escalation part, patients received 25 to 250 mg of ensartinib. In the dose-expansion part, the rate of response (RR) was 60%, the DoR was 12.8 months, and the mPFS was 9.2 months. Ensartinib was active regardless of previous treatments with TKIs and the presence of CNS metastases [44]. Additionally, the phase II part evaluated 60 patients with advanced ALK-rearranged NSCLC. The DCR was 81.7% with an ORR of 60% and an mPFS of 9.2 months. In TKI-naive patients, ORR and mPFS were 80% and 26.2 months, respectively; in those progressing to crizotinib, ORR was 69% and mPFS was 9.0 months; and in patients progressing after 2 lines of TKIs, mPFS was 1.9 months [45]. More recently, the phase 3 multicentered, randomized eXalt3 study was presented by Horn et al. at the International Association for the Study of Lung Cancer World Conference on Lung Cancer Virtual Presidential Symposium. Horn and her colleagues randomized 290 patients with ALK-positive (confirmed by central Abbott FISH test) NSCLC to either ensartinib or crizotinib. At the data cutoff, based on a pre-planned interim analysis design, there were 139 patients with progressing disease. The median PFS with ensartinib was 25.8 months compared with 12.7 months with crizotinib (HR 0.52; 95% CI 0.36–0.75). Moreover, the overall response rate was 75% vs. 67% with the first-generation ALK inhibitor. This remarkable achievement demonstrated that ensartinib could represent a new option in the first-line setting ALK-positive NSCLC disease [46].

### 2.7. Entrectinib

Entrectinib is a potent, selective, oral inhibitor of TRKA, TRKB, TRKC, ROS1, and ALK, with the ability to cross the blood–brain barrier (BBB) in preclinical models and a strong intracranial activity. This novel ALK-TKI was evaluated in two phase I/II basket trials, the Alka-372-001 trial and the STARTRK-1, enrolling solid tumors with rearrangements of the TRK family, ROS1, or ALK. In the two trials, 27 patients with ALK-rearranged solid tumors were present. Among the 19 patients pretreated with one or more ALK TKIs, no response to entrectinib was recorded. In the remaining 7 TKI-naive patients, the ORR was 57% (95% CI 25–84%) and responses were observed in ALK-rearranged NSCLC, renal cell carcinoma, and colorectal cancer. The median DoR in the ALK population was 7.4 months (95% CI 3.7 months—not reached) with an mPFS of 8.3 months (95% CI 4.6–12 months); mOS was not achieved (95% CI 19 months—not reached) and the rate of patients surviving at one year was 89.4% (95% CI 75.5–100%) [47,48].

## 3. Intracranial Efficacy

Approximately 40% of NSCLC patients present CNS involvement during the disease course along with worsening prognosis and quality of life [49]. A total of 20–30% of ALK-positive patients present with CNS metastases at the time of diagnosis but the risk increases up to 50% throughout the disease (reaching 58% at 3 years) [49]. The common treatments include local therapy such as surgery, stereotactic radiosurgery, and whole-brain radiotherapy [50]. However, the development of new targeted agents is changing the treatment approach and may represent an important turning point in the management of brain metastases (BM). The effectiveness of ALK inhibitors on the CNS depends on several factors and it seems to be related to both the tumor molecular characteristics and the drug pharmacokinetic features. Indeed, according to a retrospective analysis of the PROFILE 1005 and 1007 trials, 70% of the patients who progressed to crizotinib presented with CNS metastasis, thus representing the most common site of progression disease (PD) [51,52]. Since crizotinib is a substrate of p-glycoprotein, it is characterized by a poor BBB penetration with low cerebrospinal fluid (CSF) concentrations and a low CSF-to-plasma ratio, which hamper the achievement of a therapeutic concentration into the brain, leading to a pharmacological resistance [2]. Despite the aforementioned issues, a pooled analysis demonstrated crizotinib CNS efficacy with an intracranial response of 18% in patients who had previously received radiotherapy and 33% in patients who had not received prior radiotherapy [51,52]. Likewise, it demonstrated a prolongation of the median time to intracranial progression (13.2 vs. 7.0 months) and a similar intracranial disease control rate (DCR) at 12 weeks in these two groups (62% and 56%, respectively) [51,52]. The effectiveness of crizotinib on BM was further supported by the PROFILE 1014 trial where 23% of patients with treated BM at baseline showed longer PFS (9.0 vs. 4.0 months; HR 0.40, 95% CI 0.23–0.69) and a better RR (77% vs. 28%) with crizotinib [53]. Ceritinib is 20 times as potent as crizotinib and it has significant activity on CNS metastasis both in patients who progressed on crizotinib and in naïve patients. As well as crizotinib, ceritinib is a substrate of pump efflux transporters; however, in vivo ceritinib showed a higher efficacy against ALK-rearranged cells and a higher lipophilicity that may allow for the molecule to diffuse through the BBB at a significant rate [54]. Clinical trials from the ASCEND program (ASCEND-1 to 5) reported intracranial responses in patients with measurable baseline brain lesions. Particularly, the phase II ASCEND-2 and -3 trials evaluated ceritinib in both crizotinib-pretreated (ASCEND-2) and crizotinib-naive (ASCEND-3) chemo-pretreated patients demonstrating a remarkable intracranial DCR of 80% [55]. Of note, a recent analysis of the ASCEND-3 confirmed the activity of ceritinib on BM with a median OS of 36.2 months (95% CI 17.7 to not evaluable) in patients with BMs at the baseline, and 55.3 months (95% CI 50.1–55.3) in patients without baseline BMs [56]. Interestingly, the phase II ASCEND-7 study evaluated the activity of ceritinib in patients with ALK and NSCLC metastatic to the brain or leptomeninges, demonstrating a durable intracranial response across all study arms regardless of prior treatments [57]. Unlike crizotinib and ceritinib, alectinib is not a substrate of p-glycoprotein [58]. As demonstrated in preclinical studies, it achieves a high CNS penetration in intracranial metastases, with a high brain-to-plasma concentration ratio [59,60]. In vivo data were confirmed in the phase I/II studies. In particular, the results from the American part of the AF-002JP study showed a remarkable CNS ORR of 75% with a CNS DCR of 100% [24,61,62]. Interestingly, a pooled analysis of CNS response to alectinib showed an outstanding intracranial ORR of 64% (95% CI 49.2–77.1) [63]. Further, patients without CNS involvement at baseline presented low incidence of progression in the CNS, underlying the impressive activity of alectinib and suggesting a preventing role [62]. Finally, alectinib efficacy against CNS metastases was supported by data from phase III studies, which demonstrated the efficacy of alectinib on CNS metastases in comparison with chemotherapy and with crizotinib [27,64,65]. In the ALUR trial, the intracranial ORR was 54.2% vs. 0% for alectinib and chemotherapy, respectively. Data from the specific analysis of alectinib CNS efficacy in the J-ALEX study suggested the ability of alectinib to reduce the risk of CNS progression in comparison with crizotinib, both in patients with baseline CNS metastases (HR 0.51; 95% CI 0.16–1.64) and in patients who did not have baseline CNS metastases (HR 0.19; 95% CI 0.07–0.53) [66]. The results strongly suggest that alectinib in patients with asymptomatic BM may delay or reduce the use of local treatments [27].

However, the intracranial efficacy of brigatinib compares favorably with other second-generation ALK TKIs [61]. Brigatinib demonstrated a superior intracranial efficacy in comparison to crizotinib in the phase 3 ALTA study, which reported an intracranial response among patients with measurable lesions of 78% and 29% for brigatinib and crizotinib, respectively [34,67].

Lorlatinib is a brain-penetrant next-generation ALK TKI, active against most known resistance mutations [61]. In the phase I trial, Shaw et al. demonstrated that lorlatinib has both systemic and intracranial activity even in TKI pre-treated patients. The phase II trial enrolled a similar population in six different expansion cohorts according to previous treatments and the status of molecular drivers [40,68]. The study confirmed a substantial intracranial efficacy ranging from 42% to 75% in patients with advanced ALK-positive disease. Data from the phase III CROWN trial were recently presented at ESMO 2020. Indeed, the numerical best overall response (BOR) of lorlatinib over crizotinib was also demonstrated in the 30 patients who had measurable BM: 14 out of 17 patients (82%) who received lorlatinib had a CR (*n* = 12) or a PR (*n* = 2) compared with 3 out of 13 (23%) patients (1 CR and 2 PR) treated with crizotinib.

Regarding the efficacy on leptomeningeal metastases, case series described the rapid radiological and clinical cerebral response to lorlatinib in patients who had leptomeningeal PD on prior ALK inhibitors [51].

Ensartinib data on CNS metastasis are scarce. The results from a multicenter phase I/II, which enrolled patients with asymptomatic CNS metastases who were ALK TKI-naïve or had received prior treatments (chemotherapy or a second-generation ALK TKI), showed CNS responses in both naïve and pretreated populations. The IRR was good in patients with baseline target CNS lesions (69%) as well as in the patients with only non-target baseline lesions (1 CR and 8 SD). The median duration of intracranial response in patients who responded was 5.8 months, with the longest duration being 24 months [44]. Table 2 summarizes the intracranial efficacy of different ALK inhibitors.

## 4. Safety

The introduction of ALKis in the clinical practice is associated with new side effects, different from the conventional ones associated with chemotherapy. The most common adverse events (AEs), which usually occur during treatment with ALK inhibitors, are the gastrointestinal (GI) AEs such as nausea, vomiting, diarrhea and constipation, increasing liver enzymes and fatigue [69]. Table 3 summarizes the principal AEs observed during phase III trials in the first-line setting ALK-rearranged NSCLC. As demonstrated in the studies which compared ALK inhibitors with chemotherapy, ALKis cause severe nausea and vomiting as well as chemotherapy, but with a higher risk of all grade nausea and vomiting. The majority of AEs in patients treated with ALK TKIs are easily manageable by dose modification or discontinuation, nonetheless, some severe AEs have been reported. Regarding crizotinib, the majority of AEs reported are grades 1–2. Compared to other ALK TKIs, crizotinib has a higher incidence of visual disorders, such as diplopia, photopsia, blurred vision, visual impairment, and vitreous floaters. These visual disorders occur in 55% to 82% of patients and are almost mild or moderate. Other common AEs associated with crizotinib include GI effects, peripheral edema, dizziness, fatigue, and decreased appetite [70]. The PROFILE 1014 study described serious AEs with an incidence of 10.5%, and involved elevated aminotransferases (14%) and neutropenia (11%) [14]. However, significant differences in severe (RR: 0.97; 95% CI 0.79–1.18) and fatal (RR: 2.24; 95% CI 0.49–10.30) AEs were not detected between crizotinib monotherapy and chemotherapy. Curiously, a lower incidence of serious AEs (SAEs) was reported among Asian patients. Of note, pulmonary toxicity from crizotinib is rare, but it is a potentially life-threatening AE [70]. As already mentioned, crizotinib and alectinib were directly compared in the two phase III studies ALEX [27] and J-ALEX [65], which demonstrated a better safety profile of alectinib as compared with crizotinib [64]. In both studies, the most serious events (grade 3–5 AEs) were 41% with alectinib and 50% with crizotinib. Interestingly, in an updated analysis of the ALEX study, the safety profile of alectinib was confirmed to be favorably compared with crizotinib, despite the longer treatment duration. Additionally, the incidence of grade 3 to 5 AEs and fatal AEs was lower with alectinib than with crizotinib (44.7% vs. 51.0% and 3.9% vs. 4.6%, respectively). In addition, fewer patients had AEs which lead to dose reduction (16.4% with alectinib vs. 20.5% with crizotinib) or discontinuation (22.4% with alectinib vs. 25.2% with crizotinib) [26]. Moreover, alectinib showed a better safety profile as compared with chemotherapy. As demonstrated in the ALUR study [64], alectinib caused fewer grade 3 adverse events which occurred in 27.1% of the patients in the arm with crizotinib and 41.2% in the chemotherapy group. In addition, treatment discontinuation due to AEs was higher in the chemotherapy group (8.8%) compared with the alectinib group (5.7%). In the crizotinib resistant patients, the findings from the pooled analysis [63] of the phase II studies showed that most AEs were grade 1 to 2 [71]; however, in 40% of patients grade 3 or higher AEs occurred, leading to treatment withdrawal or discontinuation or modification in 6–33% of patients. The most common grade 3–4 AEs were represented by the increased blood creatine phosphokinase and transaminases [24]. Alectinib is a well-tolerated ALK TKI and its most common AEs include fatigue, myalgia, and peripheral edema [72].

The safety data on ceritinib come from the ASCEND program starting from the ASCEND-1 study. In the ASCEND-1 and -2 studies, ceritinib was well tolerated and manageable with dose reduction or, rarely, temporary discontinuation. The most frequent AEs reported in both studies were GI AEs and the increasing of transaminases. It is noteworthy that patients experienced an improvement in QoL. Regarding serious AEs (grade 3 or 4), they occurred in 12% and 17% of patients in the ASCEND-1 and −2 trials, respectively, and included pneumonitis (3% of patients), diarrhea, nausea, liver function test (LFT) abnormalities, hyperglycemia, and pericarditis (each effect with a frequency of 1% of patients) [18]. Consistent with data from the above-mentioned studies, the phase III ASCEND-4, which compared ceritinib to platinum-based chemotherapy, confirmed diarrhea (85%), nausea (69%), vomiting (66%), and an increase in alanine aminotransferase (60%) as the most common AEs. Likewise, these results were obtained in the phase III trial (ASCEND-5), in which patients progressing on chemotherapy and crizotinib were randomized to receive ceritinib or chemotherapy. Of note, in this study a considerable portion of patients (43%) receiving ceritinib experienced serious AEs vs. 32% in the chemotherapy group. The most frequent severe AEs were γ glutamyl transferase concentration (21%) and increased aspartate aminotransferase (14%) in patients treated with ceritinib. Concerning brigatinib, it appears to be a broadly well-tolerated agent: the most common reported toxicities were nausea, fatigue, and diarrhea with the most frequent severe AEs being increased lipase concentration, dyspnea, and hypertension [73]. In the ALTA trial, Kim et al. demonstrated that the higher dose of brigatinib was associated with a safety profile including the common toxicities reported with other ALK inhibitors: nausea, fatigue, and diarrhea [74]. The most described grade 3–4 treatment-related AEs were increased lipase concentration (9%), dyspnea (6%), and hypertension (5%). As emerged from the phase 3 ALTA trial, brigatinib is associated with less grade 3 to 5 AEs compared with crizotinib: 61% of patients in the brigatinib group and 55% of patients in the crizotinib group. AEs occurring the most with brigatinib comprised an increased creatine kinase level (39%), a cough (25%), hypertension (23%), and an increased lipase level (19%). AEs more common with crizotinib than with brigatinib included nausea (crizotinib 56% vs. brigatinib 26%), diarrhea (55% vs. 49%), constipation (42% vs. 15%), peripheral edema (39% vs. 4%), vomiting (39% vs. 18%), a higher alanine aminotransferase level (32% vs. 19%), anorexia (20% vs. 7%), photopsia (20% vs. 1%), dysgeusia (19% vs. 4%), and visual impairment (16% vs. 0%) [67]. Even if data comparing alectinib with brigatinib are lacking, when indirectly comparing ALTA-1L and ALEX alectinib is associated with fewer grade ≥ 3 adverse events (41% alectinib vs. 61% brigatinib) [75].

It is reported that AEs associated with lorlatinib are mostly mild to moderate, and can be managed with dose modification and/or supportive medical therapy. The most frequent AEs are hypercholesterolemia (82.4%), hypertriglyceridemia (60.7%), edema (51.2%), peripheral neuropathy (43.7%), and CNS effects (39.7%) [76]. Of note, the incidence of the most common AEs experienced by pretreated patients (76%) included in the phase I/II was consistent with the ITT population: hypercholesterolemia (72%), hypertriglyceridemia (39%), peripheral neuropathy (39%), and peripheral edema (39%) [68]. The phase II study confirmed the most common AEs across all patients. Of note, CNS effects of any cause were reported in 39% of patients, including changes in cognitive function (23%), mood (22%), and speech (8%), yet they were grade 1 or 2 in severity and reversible after dose reduction [40]. It is noteworthy that, different from other ALK TKIs, hepatotoxicity was less frequent with lorlatinib. Indeed, alterations in liver function were described in up to 28% of patients, yet severe hepatotoxicity occurred only in 2% of patients. Other AEs associated with lorlatinib include GI toxicity anorexia, fatigue, edema, visual disturbances, peripheral neuropathy, and cognitive effects [77]. Results from a planned interim analysis of the CROWN trial showed that the incidence of grade 3/4 AEs was higher with lorlatinib (72.5%) than crizotinib (55.6%), with the majority of AEs in the lorlatinib arm being laboratory abnormalities. Nonetheless, patients experienced AEs leading to treatment discontinuation (6.7% vs. 9.2%). Concerning ensartinib, the safety data came from the dose-escalation part of the phase I study. Treatment was well-tolerated and the most frequently reported toxicities were rash (56%), nausea (36%), pruritus (28%), and vomiting (26%). Some differences emerged in comparison to other ALKis including a lower frequency of GI AEs: diarrhea was reported less with ensartinib as compared with brigatinib or ceritinib, and vomiting occurred frequently with ceritinib yet it was observed in only a quarter of the patients with ensartinib. Last but not least, the frequency and severity of early pulmonary toxicities that have been reported with brigatinib and ceritinib have not been observed with ensartinib [44].

## 5. Mechanisms of Resistance

Despite the clinicians’ efforts, after a median period of 10.9 months all ALK-positive patients progress due to different mechanisms of resistance, which have been classified as ALK-dependent and ALK-independent. Commonly, ALK-dependent resistances occur as a result of secondary mutations within the target kinase which block the TKI binding to the target kinase. Additionally, the main secondary resistance mutations located in the ALK tyrosine-kinase domain are the gatekeeper L1196M (present in 7% of crizotinib-resistant cases) and the G1269A mutation (4% of cases) [78]. The solvent-front G1202R mutation (2% of cases) grants high-level resistance to crizotinib, as well as to next-generation ALK inhibitors. Notably, upon progression on a second-generation ALK TKI, emerging data from studies of the third-generation lorlatinib have been promising. In fact, lorlatinib demonstrates great efficacy against different ALK-dependent resistance mechanisms including L1196M and G1202R substitutions [78].

However, ALK-independent mechanisms of resistance are amplifications of the ALK fusion gene, or alternative signaling pathways such as the amplification of epidermal growth factor receptor (EGFR) or of insulin-like growth factor (IGF-1R) or c-kit mutations; epithelial to mesenchymal transition (EMT) or change in tumor histology. Particularly, the transformation from adenocarcinoma to small-cell lung cancer has rarely been described as a mechanism of resistance. Understanding these ALK-independent mechanisms of resistance is a clinical challenge and future studies to investigate combination treatments in this subset are mandatory. In order to overtake acquired resistance to first-line ALK TKIs, several second- and third-generation ALK inhibitors have been developed in the last few years. Table 4 displays the main mechanisms of resistance to ALK-TKIs.

## 6. Sequence of Therapy

As mentioned above, novel targeted drugs are currently available and active in terms of both extra and intracranial disease. However, these drugs might be dissimilar in terms of the targeting profile, particularly against resistance mutations to previous TKIs and safety profiles, indicating that ALK-TKIs’ use may be influenced according to patients’ characteristics and the biological features of the tumor. Therefore, a comprehensive molecular characterization is mandatory to decide the best therapeutic sequence to offer patients the strongest and longest response along with the best quality of life. Recently, the treatment paradigms have shifted and second-generation ALK-TKIs are the first-line treatment of choice nowadays for advanced ALK-rearranged NSCLC. Indeed, based on the results of the J-ALEX and ALEX trials, which confirmed the safety and efficacy profile, and its activity on brain metastases, alectinib has been the preferred first-line therapy in this setting.

Notwithstanding, crizotinib remains the standard of care treatment for naïve ALK-positive patients in several regions of the world due to pharma-economic assessment. In Italy, alectinib is now the first-line preferred treatment for advanced ALK-positive NSCLC according to the AIOM guidelines [29].

In addition, as mentioned above, various ALK-resistant mutations arise after treatment with second-generation ALK inhibitors. The most frequently described ALK mutation is the solvent-front mutation G1202R, accounting for around 2% of patients treated with second-generation ALK TKIs [78]. The identification of ALK-resistant mutations determines the importance of obtaining re-biopsy after progressive disease, either by tissue or liquid samples, leading to more appropriate TKI treatment with a better understanding of resistance mechanisms. Liquid biopsy could be particularly relevant when tissue specimens for analysis are insufficient, critical delays in regular diagnosis are expected, or contraindications to the tissue biopsy exist. Moreover, liquid biopsy is recommended in the College of American Pathologists (CAP)/International Association for the Study of Lung Cancer (IASLC)/Association for Molecular Pathology (AMP) guideline for the molecular testing of patients with NSCLC [80]. Various clinical studies have been conducted evaluating the concordance rate among plasma samples and tissue specimens [81,82,83,84,85]. The overall concordance rate for ALK fusion was reported at 95.7%, with a positive predictive value of 100% [83]. These recent findings are doubtless encouraging, yet some drawbacks should be improved before the liquid detection of molecular aberrations becomes a standard tool. Until then, tissue sample analysis remains the standard of care for monitoring patients on treatment and selecting the next therapeutic options. A treatment algorithm is proposed in Figure 1. This algorithm is based on the currently available data according to NCCN, ESMO, and AIOM guidelines. Our preference is to utilize lorlatinib, a third-generation ALK-TKI, upon progression on a second-generation ALK inhibitor due to the absence of putative mutations. Accordingly, recent evidence suggests that even in ALK-positive patients not presenting with a resistance point mutation, reasonable response rates (ORR 32%) and an mPFS of 5.5 months may be attained on lorlatinib treatment [41].

Notwithstanding, more research is awaited with the aim of guiding better treatment decision-making in patients with and without ALK-resistance mutations. When patients have progressed on lorlatinib, the next-line therapy may be chemotherapy, immunotherapy, a combination of both, or participation in available clinical trials.

Of note, there are no available data of randomized clinical trials which have compared lorlatinib with the other third-generation or second-generation ALK TKIs.

Moreover, considering the currently available data and guidelines, the best therapeutic algorithm suggested for patients with asymptomatic CNS metastases at diagnosis should be an upfront systemic therapy with the next-generation ALKi due to the high intracranial activity, and postponing the use of local therapy (radiotherapy or surgery) when the patients start to show intracranial symptoms [58]. Moreover, the NCCN-NSCLC panel has deleted the option to continue crizotinib in patients with BM who had progressed after first-line therapy with crizotinib, suggesting the use of the other ALK inhibitors in this setting, as indicated in NCCN guidelines [31].

However, novel prospective studies are necessary to establish the best therapeutic strategy for patients with CNS disease, and the management of CNS disease should always be discussed in a multidisciplinary team. Notably, in this setting recent studies are focusing on the CSF liquid biopsy as a useful tool in case of massive and rapid progression in order to redefine the biology of the tumor and to overcome the resistance by the switch to another adequate ALK inhibitor [51,86,87,88,89,90,91].

## 7. Future Perspectives

The therapeutic and clinical landscape of ALK-positive NSCLCs is rapidly evolving, as second-generation ALK inhibitors have nowadays become the well-established first-line treatment in this setting. Moreover, a better understanding of the optimal treatment sequence is awaited, since modern and more accurate detection techniques of molecular biology keep improving. Particularly, NGS empowers the analysis of various biomarkers for different patients at the same time [92]. However, several questions are still pending and the mechanisms of resistance to TKI should be more thoroughly understood. Additionally, as mentioned above, the optimal sequential employment of next-generation ALK inhibitors is crucial and perhaps the key to successful therapeutic strategy. However, further clinical investigations are needed to improve the therapeutic management in this subgroup of patients and produce more robust efficacy data. Recently, the emergence of immune checkpoint inhibitors, which alter tumoral immunosuppression and enhance immunity, has reformulated the clinical management of patients with advanced NSCLC [93].

Several authors have recently demonstrated that up to 60% of cases of ALK-positive NSCLC tumors expressed a significant level of PD-L1 compared to EGFR/KRAS/ALK wild type NSCLC [94,95]. Of note, Hong et al. successfully showed in in vitro assays that anti-PD-1/PD-L1 antibodies could potentially be therapeutic in ALK-positive NSCLC [96]. Yet, the potential use of anti-PD-L1 therapy in this subset of patients is not supported by isolated case reports and retrospective analysis, in which ALK-positive NSCLC patients had low response rates to PD-1/PD-L1 inhibitors [97]. Thus, current efforts are focusing on new potential combination strategies with antitumor activity, using immune checkpoint blockade and ALK-TKIs [98,99,100]. In fact, new evidence aiming to support novel treatments in ALK-positive NSCLC patients is awaited from prospective clinical trials, which are evaluating the safety and efficacy of PD-1/PD-L1, CTLA-4 and target agents with ALK inhibitors [101] (a summary of ongoing clinical trials is presented in Table 5).

Moreover, the benefit of chemotherapy after the failure of ALKi-based therapy remains limited; however, standard options include platinum/pemetrexed-based chemotherapy treatment. Notably, a recent multicenter retrospective clinical trial conducted by Lin et al. evaluated the efficacy of such chemotherapy regimens in patients refractory to second-generation TKIs. The ORR of platinum/pemetrexed-based chemotherapy was 29.7% (95% CI 15.9–47.0%) [102].

Furthermore, the strategies for combination chemo-immunotherapy have changed the treatment landscape of NSCLC. Notably, in the open-label phase III IMpower150 study, which evaluated the combination of atezolizumab, bevacizumab, and chemotherapy in patients with metastatic non-squamous NSCLC who had not previously received chemotherapy, patients with ALK genomic alterations were included. Authors randomly assigned patients to receive atezolizumab, carboplatin, and paclitaxel (ACP); bevacizumab, carboplatin, and paclitaxel (BCP); or atezolizumab and BCP (ABCP) every 3 weeks for 4 or 6 cycles, followed by maintenance therapy. Surprisingly, the PFS among patients with ALK aberrations was longer with ABCP than with BCP (9.7 mo. vs. 6.1 mo., HR 0.59; 95% CI 0.37–0.94) [103]. These findings in a subset of patients with limited treatment options are particularly relevant and warrant further investigation (Table 5).

## 8. Conclusions

Over the last few decades, novel and more selective compounds directed to ALK-rearranged NSCLCs were conceived with better activity, both in terms of extra and intracranial disease, higher specificity, and targeting an extensive subset of resistance mutations. Of note, ALK inhibitors differ for the targeting profile, particularly against the mechanism of resistance mutations, indicating that the therapeutic choices might be influenced and guided by the patients’ characteristics.

Therefore, a greater interpretation of the biological evolution of the tumor and the several intrinsic molecular features is mandatory. Furthermore, to define the best strategy of therapeutic sequence, a better knowledge of the biological and clinical aspects of cancers harboring ALK aberrations is highly recommended, thus offering patients the strongest and longest response with the best quality of life.

## Figures and Tables

**Figure 1 pharmaceuticals-13-00474-f001:**
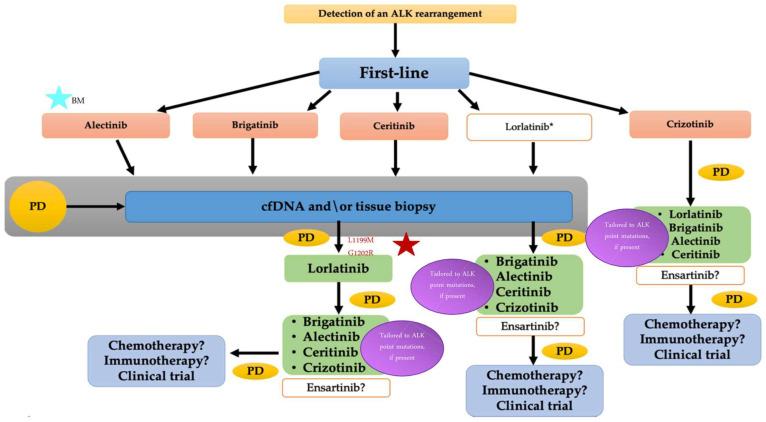
Proposed treatment algorithm for ALK-rearranged advanced NSCLC. Abbreviations: ALK, anaplastic lymphoma kinase; cfDNA, circulating-free DNA; PD, progressive disease; BM, brain metastases.

**Table 1 pharmaceuticals-13-00474-t001:** Main phase III trials evaluating anaplastic lymphoma kinase (ALK)-inhibitors in the first-line setting ALK-positive non-small cell lung cancer (NSCLC).

Clinical Trial	Drugs	PFS	Primary Endpoint
PROFILE 1014	Crizotinib vs. platinum-base CT	10.9 vs. 7 mo. (HR 0.45; 95% CI 0.35–0.60)	PFS
ASCEND-4	Ceritinib vs. platinum-based CT	16.6 vs. 8.1 mo. (HR 0.55; 95% CI 0.42–0.73)	PFS
J-ALEX	Alectinib vs. crizotinib *	34.1 vs. 10.2 mo. (HR 0.37; 95% CI 0.26–0.52)	PFS
ALEX	Alectinib vs. crizotinib **	34.8 vs. 10.9 mo. (HR 0.43; 95% CI 0.32–0.58)	PFS
ALESIA	Alectinib vs. crizotinib ***	NE vs. 11.1 mo. (HR 0.22; 95%CI 0.13–0.38)	PFS
ALTA 1L	Brigatinib vs. crizotinib	24 vs. 11 mo. (HR 0.49; 95% CI 0.33−0.74)	PFS
eXalt3	Ensartinib vs. crizotinib	25.8 vs. 12.7 mo. (HR 0.52; 95% CI 0.36–0.75)	PFS
CROWN	Lorlatinib vs. crizotinib	NE vs. 9.3 mo. (HR 0.21; 95% CI 0.14–0.30)	PFS

Abbreviations: PFS, progression-free survival; OS, overall survival; RR, response rate; CT, chemotherapy; mo., months; HR, hazard ratio; CI confidence interval; * J-ALEX included only the Asian population and alectinib was administered at a dose of 300 mg orally twice daily; ** ALEX study included only the Caucasian population; *** In the ALESIA trial, alectinib was administered at a dose of 600 mg orally twice daily.

**Table 2 pharmaceuticals-13-00474-t002:** Summary of ALK inhibitors’ efficacy for brain metastases in ALK-positive treatment of naive NSCLC.

Clinical Trial	Drugs/Phase	No. of pts w/BM	PFS	OS	IRR	IORR	IDCR	IDOR
PROFILE 1005	Crizotinib, 2	166	8.4 mo.	21.8 mo.	NA	33%	62%	NR
PROFILE 1007	Crizotinib, 3	109	7.7 mo.	12.2 mo.	NA	18%	56%	NR
PROFILE 1014	Crizotinib, 3	39	9.0 mo.	17.4 mo	77%	15%	NA	NR
ASCEND-1	Ceritinib, 1	94	18.4 mo.	NR	NA	72%	79%	NA
ASCEND-2	Ceritinib, 2	100	5.7 mo.	NR	NA	45%	80%	NA
ASCEND-3	Ceritinib, 2	50	10.8 mo.	36.2 mo.	NA	20%	80%	9.1 mo.
ASCEND-4	Ceritinib, 3	54	16.6 mo.	NR	NA	73%	NA	16.6 mo.
ASCEND-5	Ceritinib, 3	66	4.4 mo.	NA	NA	35%	NA	6.9 mo.
ASCEND-6	Ceritinib,1/2	103	5.7 mo.	NA	NA	39.1%	82.6%	NA
ASCEND-7	Ceritinib, 2	138	5.4 mo.	NA	NA	51.5%	75.8%	7.5 mo.
ALUR	Alectinib, 3	72	7.1 mo.	NA	NA	54.2%	NA	NR
ALEX	Alectinib, 3	64	34.8 mo.	48.2 mo	59%	81%	NA	17.3 mo.
ALTA	Brigatinib, 3	41	29.4 mo.	NA	NA	78%	NR	NA
CROWN	Lorlatinib, 3	30	18.3 mo.	NR	NA	76%	NA	NE
NCT01625234	Ensartinib, 1/2	35	9.2 mo.	NA	69%	64.3%	NA	5.8 mo.

Abbreviations: No. of pts w/ BM, number of patients with brain metastases; IORR, intracranial objective response rate; IDOR, intracranial duration of response; NR, not reported; mo., months; NA, not available; PFS, progression-free survival; OS, overall survival; IRR, intracranial response rate; IDCR, intracranial disease control rate.

**Table 3 pharmaceuticals-13-00474-t003:** Summary of principal adverse events (AEs) observed during phase III trials in the first-line setting ALK-positive NSCLC.

	PROFILE 1014	ASCEND4	ALEX	ALTA-1L	CROWN	eXalt3
Grade 3/4 (%)						
Any AEs	-	78	41	61	-	23
Diarrhea	1	5	0	1	-	0
Nausea	1	3	1	1	0	1
Anorexia	30	29	-	-	-	1
Weight reduction	-	4	-	1	-	-
Vomiting	2	5	5	0	-	1
Constipation	2	0	0	0	0	-
Blood creatinine increased	-	2	2	0	-	-
Peripheral edema	1	-	-	1	0	-
Rash	-	-	-	0	-	12
Dizziness	0	-	0	-	-	-
Visual impairment	1	-	0	-	0	-
Asthenia	0	3	-	-	-	-
AST and/or ALT increased	14	31	5	1	2	-
Bilirubin increased	-	-	2	-	-	-
Γ-glutamyltransferase increased	-	29	1	-	-	-
Myalgia	-	-	0	-	-	-
Headache	1	0	-	-	-	-
QT interval prolongation	-	-	-	-	-	-
Alkaline phosphatase increased	-	7	-	-	-	-
Neuropathy	1	-	-	-	-	-
Neutropenia	11	1	-	0	-	-
Anemia	0	2	5	0	-	-
Stomatitis	1	-	-	-	-	-
Treatment discontinuation due to AEs	12	5	11	12	0	5.2
Interstitial lung disease	-	0	-	-	-	-
Pruritus	-	-	-	1	0	5
Cognitive effects	-	-	-	-	2	-
Hypertriglyceridemia	-	-	-	-	0	-
Hypercholesterolemia	-	-	-	-	-	-
Hypertension	-	-	-	10	-	-

Abbreviations: AE, adverse event; AST, aspartate aminotransferase; ALT, alanine aminotransferase.

**Table 4 pharmaceuticals-13-00474-t004:** Principal mechanisms of resistance to ALK inhibitors.

	ALK-Independent Resistance Mechanism	ALK-Dependent Resistance Mechanism
Crizotinib	EGFR overexpression and IGF-1R activation	Amplification of the ALK fusion gene; L1196M, G1269A/S, I1151Tins, L1152P/R, C1156Y/T, I1171T/N/S, F1174C/L/V, V1180L, G1202R, S1206C/Y, E1210K mutation acquisition
Ceritinib	c-Met gene amplification; activating mutation of MEK and PIK3CA mutations	G1202R, F1174C/L/V, G1202del, I1151Tins, L1152P/R, C1156Y/T
Alectinib	c-Met gene amplification and PIK3CA mutations	G1202R, I1171T/N/S, V1180L, L1196M
Brigatinib	Not reported	E1210K + S1206C, E1210K + D1203N, G1202Ra
Lorlatinib	NF2 loss of function mutations	L1198F + C1156Yc, L1196M/D1203N, F1174L/G1202R, C1156Y/G1269A [79]

Abbreviations: ALK, anaplastic lymphoma kinase; IGF-1R, insulin growth factor-1 receptor; EGFR, epidermal growth factor receptor.

**Table 5 pharmaceuticals-13-00474-t005:** Summary of ongoing trials with combination therapy in ALK-rearranged NSCLCs (source: clinicaltrials.gov).

Clinical Trial Identifier	Study Title	Drug	Phase	Status
**NCT02393625**	Study of Safety and Efficacy of Ceritinib in Combination with Nivolumab in Patients With ALK-positive Non-small Cell Lung Cancer	Ceritinib + Nivolumab	I	Active, not recruiting
**NCT03087448**	Ceritinib + Trametinib in Patients with Advanced ALK-Positive Non-Small Cell Lung Cancer	Ceritinib + Trametinib	I/II	Recruiting
**NCT04227028**	Brigatinib and Bevacizumab for the Treatment of ALK-Rearranged Locally Advanced, Metastatic, or Recurrent Non-small Cell Lung Cancer	Brigatinib + Bevacizumab	I	Recruiting
**NCT03202940**	A Phase IB/II Study of Alectinib Combined with Cobimetinib in Advanced ALK-Rearranged (ALK+) NSCLC	Alectinib + Cobimetinib	IB/II	Recruiting
**NCT02521051**	Phase I/II Trial of Alectinib and Bevacizumab in Patients with Advanced, Anaplastic Lymphoma Kinase (ALK)-Positive, Non-Small Cell Lung Cancer	Alectinib + Bevacizumab	I/II	Recruiting
**NCT01998126**	Combination Checkpoint Inhibitor Plus Erlotinib or Crizotinib for EGFR or ALK Mutated Stage IV Non-Small Cell Lung Cancer	ICI + Erlotinib/Crizotinib	I	Recruiting
**NCT04292119**	Lorlatinib Combinations in Lung Cancer	Lorlatinib Crizotinib Binimetinib	I/II	Recruiting
**NCT03611738**	Ceritinib Plus Docetaxel in ALK-Negative, EGFR WT Advanced NSCLC	Ceritinib + Docetaxel	I	Recruiting
**NCT04005144**	Brigatinib and Binimetinib in Treating Patients with Stage IIIB-IV ALK or ROS1-Rearranged Non-small Cell Lung Cancer	Brigatinib + Binimetinib	I	Recruiting
**NCT03202940**	A Phase IB/II Study of Alectinib Combined with Cobimetinib in Advanced ALK-Rearranged (ALK+) NSCLC	Alectinib + Cobimetinib	IB/II	Recruiting
**NCT02521051**	Phase I/II Trial of Alectinib and Bevacizumab in Patients with Advanced, Anaplastic Lymphoma Kinase (ALK)-Positive, Non-Small Cell Lung Cancer	Alectinib + Bevacizumab	I/II	Recruiting

Glossary: ICI, Immune-Checkpoint Inhibitor; WT, Wild Type; ALK, Anaplastic Lymphoma Kinase; EGFR, Epidermal Growth Factor Receptor.

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
