# Peer review of "The Emerging Therapeutic Landscape of ALK Inhibitors in Non-Small Cell Lung Cancer"

_pharmaceuticals, 2020, doi:10.3390/ph13120474_

Round 1

Reviewer 1 Report

This review is aimed at covering the field of tyrosine kinase inhibition with reference to the current approved ALK inhibitors. The English standard could be improved with inconsistent use of grammar prevalent throughout the document. The paper begins with an overview of each compound which is relatively well thought out and nicely referenced. Some grammatical errors need the attention of a native English speaker in this section.

The problems with this paper begin with the section Intracranial efficacy through to the end. No attempt has been made to summarize the data in a tabulated form and the section consists of one paragraph two and half pages long. It is almost impossible to read this section with any understanding since the information appears to have been copied directly from individual papers and spliced together, boarderline plagerism! Sentence appear almost exactly the same between this paper and the referenced papers. With some sections of data no references are given, for example page 8 contains just 3 references for a substantial range of information, (note: I did check with each and everyone of the quoted references). It does not appear if the author has made any attempt to summarized data across different references to build a picture of the current knowledge and present it in a clear fashioned. This is the lazy approach to a review and one I do not agree with.

Author Response

Reviewers' comments:

Reviewer #1: This review is aimed at covering the field of tyrosine kinase inhibition with reference to the current approved ALK inhibitors. The English standard could be improved with inconsistent use of grammar prevalent throughout the document. The paper begins with an overview of each compound which is relatively well thought out and nicely referenced. Some grammatical errors need the attention of a native English speaker in this section.

The problems with this paper begin with the section Intracranial efficacy through to the end. No attempt has been made to summarize the data in a tabulated form and the section consists of one paragraph two and half pages long. It is almost impossible to read this section with any understanding since the information appears to have been copied directly from individual papers and spliced together, boarderline plagerism! Sentence appear almost exactly the same between this paper and the referenced papers. With some sections of data no references are given, for example page 8 contains just 3 references for a substantial range of information, (note: I did check with each and everyone of the quoted references). It does not appear if the author has made any attempt to summarized data across different references to build a picture of the current knowledge and present it in a clear fashioned. This is the lazy approach to a review and one I do not agree with.

Authors’ reply

1/ No attempt has been made to summarize the data in a tabulated form

R: Thank you. We provided a new table containing all trials discussed in this paragraph.

2/ the section consists of one paragraph two and half pages long. It is almost impossible to read this section with any understanding

R: Thank you for your precious comment. We summarized these pages trying to focus on the most relevant information.

3/ the information appears to have been copied directly from individual papers and spliced together, borderline plagiarism!

R: Thank you again. We have performed a re-evaluation of the entire manuscript, trying to rewrite the parts most similar to the individual cited papers. Then, we performed a formal check to identify the sources of plagiarism with a specialized software.

4/ With some sections of data no references are given, for example page 8 contains just 3 references for a substantial range of information

R: Thank you. We evaluated the entire manuscript by adding the references necessary to justify the sources of each information.

5/ Some grammatical errors need the attention of a native English speaker in this section.

R: Thank you. We thoroughly performed a scientific English language revision and spell checking.

Reviewer 2 Report

The review is aimed to describe the modern therapeutic strategies for the patients with Non-Small cell lung cancer (NSCLC) harboring activating ALK mutations. In general, the manuscript is very-well written and provides the detailed analysis of the ALK inhibitors that are currently used and/or considered as a perspective for therapy of NSCLC.

Indeed, introduction of crizotinib to the clinical oncology dramatically improved the prognosis of ALK-positive NSCLC patients when compared to the standard chemotherapies. However, despite an impressive response rates the tumors eventually developed the resistance to the targeted therapy, thereby illustrating a rationale for developing of the second-line therapies. The manuscript describes the mechanisms of resistance that are composed of ALK-dependent and independent mechanisms. These mechanisms are well-described and shown in a separate chapter (# 5). All the tables shown in the manuscript are very illustrative, informative and helpful. 

Minor:

Figure 1 do not provide the clear algorithm for therapy of ALK-rearranged NSCLC and the direction to make a choice to use the specific TKi. I suggest to add the pattern of ALK mutations into the first line of therapy to illustrate the specific benefits to use each type of TKi.

Moreover, (to my opinion), the benefits of chemotherapy for ALK-positive NSCLC after the failure of targeted-based therapies remain illusive (the info presented in Table 1 supports this point).  

Similarly, no data is available in the manuscript about the potential benefits of  immunotherapy for the patients with NCLSC (as an option for the third-line therapy after TKi failure). The data about PDL1/PD-1 expression in this subset of patients will be very interesing (if it is available) and provide a rationale to use immune checkpoint inhibitors. Otherwise, it looks like as a preliminary idea of the authors which is not supported by experimental and clinical data. By the way, the authors are mentioning about this (lines 675-680), and also refering to the Table 5. However, no data and references about the ongoing clinical trials to examine the efficiency of check-point inhibitors for ALK-positive NSCLC are shown in the manuscript.    

Author Response

Reviewer #2: The review is aimed to describe the modern therapeutic strategies for the patients with Non-Small cell lung cancer (NSCLC) harboring activating ALK mutations. In general, the manuscript is very-well written and provides the detailed analysis of the ALK inhibitors that are currently used and/or considered as a perspective for therapy of NSCLC.

Indeed, introduction of crizotinib to the clinical oncology dramatically improved the prognosis of ALK-positive NSCLC patients when compared to the standard chemotherapies. However, despite an impressive response rates the tumors eventually developed the resistance to the targeted therapy, thereby illustrating a rationale for developing of the second-line therapies. The manuscript describes the mechanisms of resistance that are composed of ALK-dependent and independent mechanisms. These mechanisms are well-described and shown in a separate chapter (# 5). All the tables shown in the manuscript are very illustrative, informative and helpful. 

Minor:

Figure 1 do not provide the clear algorithm for therapy of ALK-rearranged NSCLC and the direction to make a choice to use the specific TKi. I suggest to add the pattern of ALK mutations into the first line of therapy to illustrate the specific benefits to use each type of TKi.

Moreover, (to my opinion), the benefits of chemotherapy for ALK-positive NSCLC after the failure of targeted-based therapies remain illusive (the info presented in Table 1 supports this point).  

Similarly, no data is available in the manuscript about the potential benefits of  immunotherapy for the patients with NCLSC (as an option for the third-line therapy after TKi failure). The data about PDL1/PD-1 expression in this subset of patients will be very interesing (if it is available) and provide a rationale to use immune checkpoint inhibitors. Otherwise, it looks like as a preliminary idea of the authors which is not supported by experimental and clinical data. By the way, the authors are mentioning about this (lines 675-680), and also refering to the Table 5. However, no data and references about the ongoing clinical trials to examine the efficiency of check-point inhibitors for ALK-positive NSCLC are shown in the manuscript.    

1/ The manuscript describes the mechanisms of resistance that are composed of ALK-dependent and independent mechanisms. These mechanisms are well-described and shown in a separate chapter (# 5). All the tables shown in the manuscript are very illustrative, informative and helpful.

R: Thank you.

2/ Figure 1 does not provide the clear algorithm for therapy of ALK-rearranged NSCLC and the direction to make a choice to use the specific TKi. I suggest to add the pattern of ALK mutations into the first line of therapy to illustrate the specific benefits to use each type of TKi.

R: Thank you for your comment. We have modified figure 1 as per your indication, even if so far no specific ALK alterations could guide the choice of the best TKI in the first line setting. However, some information has been added about the resistance point variants that could suggest a therapeutic option in the second-line scenario, although there are no implications in current clinical practice.

3/ Moreover, (to my opinion), the benefits of chemotherapy for ALK-positive NSCLC after the failure of targeted-based therapies remain illusive (the info presented in Table 1 supports this point). 

R: Thank you. We do agree with your comment. We specified that the benefit of chemotherapy remains on progression from TKIs.

4/ Similarly, no data is available in the manuscript about the potential benefits of immunotherapy for the patients with NCLSC (as an option for the third-line therapy after TKi failure).

R: Thank you again. We have reported preliminary data on immunotherapy. We have also included data on ongoing trials (see references from 98-105).

5/ English minor language revision is needed

R: Thank you. We revised English as you suggested.

Round 2

Reviewer 1 Report

Firstly, I would like to compliment the authors for taking the time and effort to revise their paper to such a significant extent.

The grammar has improved by an order of magnitude and the reworked sections 3,4,5,6 and 7 have made the paper much clearer. The authors have removed data from the text and incorporated clear tables. While the text could be split into more paragraphs the current improvement is still sufficient for publication. I have no further remarks to make about this paper.